# Setting benchmarks for modelling gas–surface interactions using coherent control of rotational orientation states

Yosef Alkoby[1,3], Helen Chadwick [1,3✉], Oded Godsi[2], Hamza Labiad[1], Matthew Bergin[1], Joshua T. Cantin [1], Ilya Litvin[2], Tsofar Maniv[2] & Gil Alexandrowicz [1✉]

The coherent evolution of a molecular quantum state during a molecule-surface collision is a detailed descriptor of the interaction potential which was so far inaccessible to measurements. Here we use a magnetically controlled molecular beam technique to study the collision of rotationally oriented ground state hydrogen molecules with a lithium fluoride surface. The coherent control nature of the technique allows us to measure the changes in the complex amplitudes of the rotational projection quantum states, and express them using a scattering matrix formalism. The quantum state-to-state transition probabilities we extract reveal a strong dependency of the molecule-surface interaction on the rotational orientation of the molecules, and a remarkably high probability of the collision flipping the rotational orientation. The scattering matrix we obtain from the experimental data delivers an ultra-sensitive benchmark for theory to reproduce, guiding the development of accurate theoretical models for the interaction of $H_2$ with a solid surface.

[1] Department of Chemistry, College of Science, Swansea University, Swansea SA2 8PP, UK. [2] Schulich Faculty of Chemistry, Technion Israel Institute of Technology, 32000 Technion City, Haifa, Israel. [3] These authors contributed equally: Yosef Alkoby, Helen Chadwick. ✉email: h.j.chadwick@swansea.ac.uk; g.n.alexandrowicz@swansea.ac.uk

The interaction of molecules with surfaces lies at the heart of many research fields and applications, including star formation, atmospheric chemistry and industrial heterogeneous catalysis[1–6]. Developing a predictive understanding of these processes has potentially great value, for example in designing more efficient catalysts[7,8]. However, even modelling the simplest molecule, $H_2$, with a metal surface accurately presents a significant challenge[9]. To develop accurate models, it is crucial to have results from fundamental surface-science experiments to benchmark theoretical descriptions against[10,11]. Currently, stringent tests of the approximations that are used in calculations[9,12–15] are obtained from carefully controlled quantum state-resolved gas–surface experiments, which have shown the role that translational, vibrational and rotational energy all play in determining the outcome of a gas–surface collision (see, for example, review articles[11,16–19] and references therein).

Arguably, the most sensitive benchmark for testing the accuracy of a theoretical model would be an experiment, which can follow the coherent propagation of a molecular quantum state as the molecule approaches the surface and scatters back, probing both the long and short-range interaction potential. This coherent propagation can be expressed by a scattering matrix, the elements of which express the changes in amplitude and phase for all the possible quantum state-to-state transitions[20,21]. Scattering matrices of this type have been calculated theoretically to model molecule–surface collisions[20,22], but were beyond the reach of existing state-of-the-art experimental methods. Here, we demonstrate for the first time to the best of our knowledge, an experimental determination of a scattering matrix. The access to the amplitude and phase changes of the quantum states is achieved by coherently controlling the rotational projection ($m_J$) states of ground state $H_2$ molecules before and after they collide with a lithium fluoride (LiF) surface, where $m_J$ is the quantum analogue of the orientation of the rotational plane of the molecule. The scattering matrix we obtain from our experiments allows us to confirm a previous theoretical prediction[23] that collisions of $H_2$ with LiF can change the rotational orientation of the molecule, as well as providing an extremely stringent benchmark which will guide the development of accurate theoretical models. In addition, we find that the quantum state-to-state scattering probabilities depend on the initial and final $m_J$ state showing the interaction potential depends sensitively on the rotational orientation of the molecule before and after the collision, and that the collisions rotationally polarise the scattered $H_2$.

## Results and discussion

**Molecular beam propagation**. The experiments we performed use magnetic fields for both particle deflection and coherent wave function control. Figure 1 shows a schematic of the experimental approach, the basic elements of which have been described in a previous publication[24]. Here, we will discuss the key aspects of the experimental method and emphasise the differences in the present study that make it possible to extract state-to-state scattering probabilities.

A molecular beam is formed by a supersonic expansion of a pure $H_2$ beam through a cold (100 K) nozzle. The cold beam is a mixture of the two lowest rotational states, the $J = 0$ singlet para-hydrogen state, and the $J = 1$ triplet of ortho-hydrogen. The former is not affected by magnetic fields and provides a constant background in our experiments while the latter splits within a magnetic field into nine different quantum states that can be represented by the nine combinations of the nuclear spin projection $m_I$ and the rotational projection, $m_J$ quantum states. Figure 2a shows the magnetic field dependence of the energies

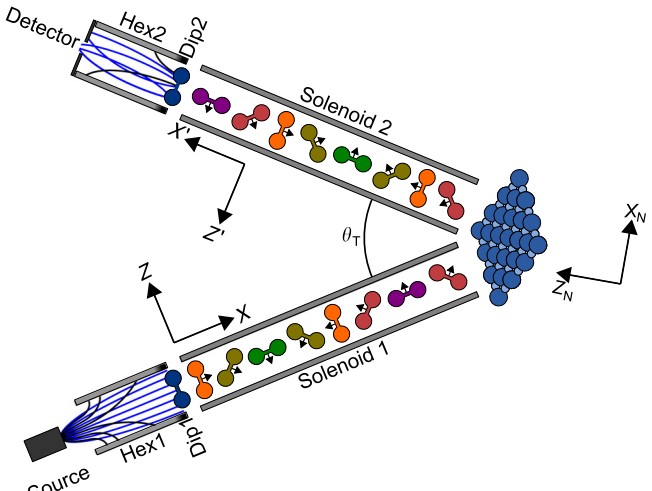

**Fig. 1 A schematic of the coherent magnetic manipulation approach[24].** Schematic of the experimental apparatus showing the position and function of the main magnetic elements as well as the different axis definitions mentioned in the text. The two hexapole fields (Hex1 and Hex2) are characterised by strong magnetic field gradients, leading to trajectory bending which either focuses (blue) or defocuses (black) the molecules depending on their $m_I$, $m_J$ state with respect to the dipole fields (Dip1 and Dip2). The hexapole fields are used for polarising and analysing the incoming and outgoing molecules respectively. In between these regions, the wave functions evolve coherently, allowing us to control and encode the rotational projection states by altering the magnetic fields before and after scattering. This is illustrated graphically in the figure as a change of the rotational plane of the propagating molecules, with the different coloured molecules representing different rotational projection states. The angle between the two arms is denoted $\theta_T$.

of these states, determined by the pioneering experiments of Ramsey[25].

The beam is passed through a magnetic hexapole field (Hex1), characterised by very strong magnetic field gradients[26,27]. Strong magnetic and electric field gradients, offer a well-known method for separating particles with different magnetic or electronic quantum states by selective deflection of their trajectories[28]. For $O_2$, which is a paramagnetic molecule, passing the beam through a magnetic field gradient is all that is required to enhance one of the rotational projection states and perform alignment sensitive scattering[29] and reactivity measurements[30]. However, this approach cannot be used for the more general case of ground state closed shell molecules such as $H_2$. This can be readily understood by looking at the magnetic field dependence of the states plotted in Fig. 2a. The states divide into three main branches depending on their nuclear spin projection and a secondary, much subtler, threefold splitting related to the different rotational projection states with the same nuclear spin projection. This type of splitting reflects the particularly weak rotational magnetic moment, and would make it extremely difficult to efficiently separate the trajectories of a particular $m_I$, $m_J$ state of $H_2$. A further more fundamental difficulty can be seen from the magnetic field energy dependence of the states. The energies of the states have a non-linear dependence on magnetic field that results in the lines crossing each other. This behaviour reflects the non-negligible coupling between the nuclear and rotational magnetic moments. The coupling means $m_I$ and $m_J$ are not eigenstates of the system, and even if an initial $m_I$ and $m_J$ state was selected using magnetic field deflection, it would mix into a superposition state within micro-seconds unless a sufficiently strong magnetic field is maintained[31].

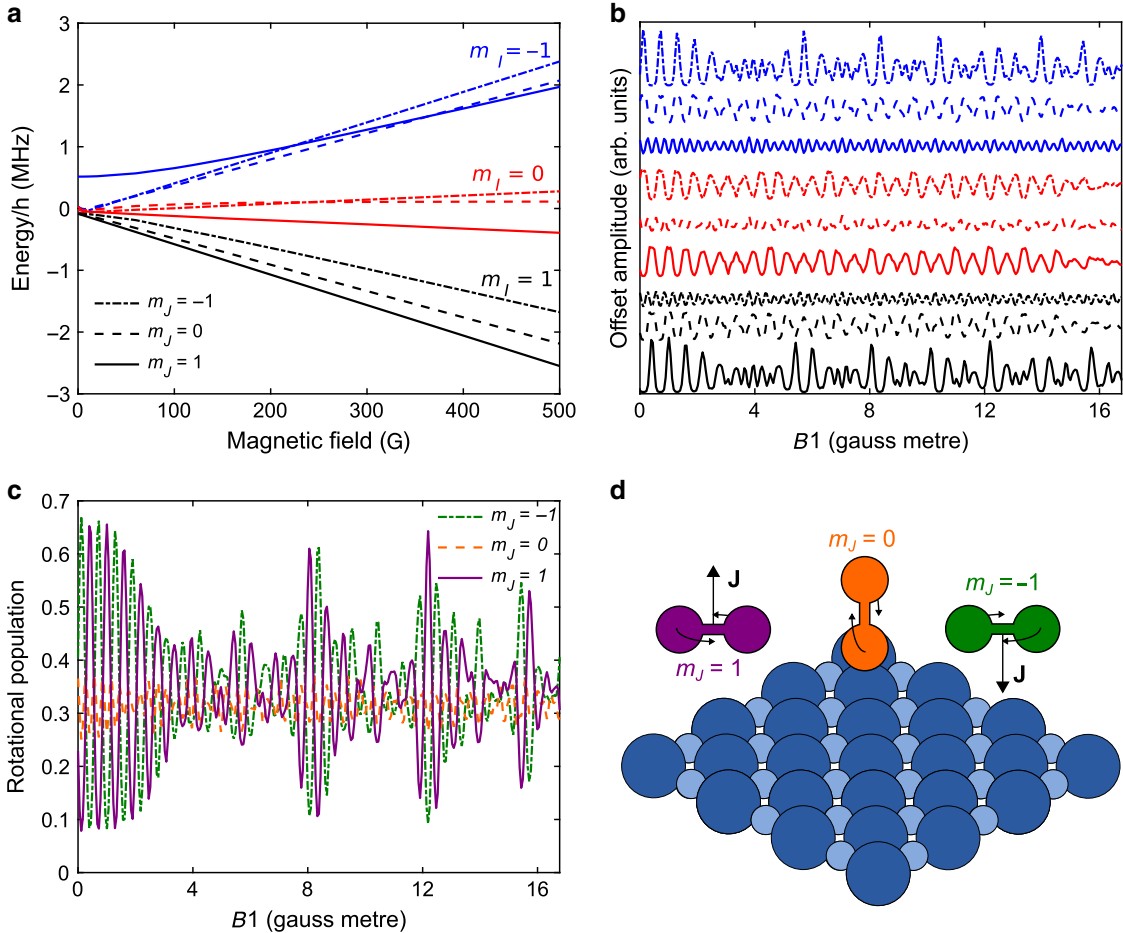

**Fig. 2 The effect of magnetic fields on ortho-hydrogen. a** The magnetic field dependence of the energy of the $m_I$, $m_J$ states of ortho-H$_2$ in $J = 1$ calculated using the Ramsey Hamiltonian[25]. **b** Calculated populations of the nine different states which reach the surface as a function of the applied magnetic field integral in the first electromagnet, $B1$. The projections are defined using the surface normal as the quantisation axis with each line being defined in the same way as in panel a, and the lines are shifted with respect to each other for clarity. **c** Calculated populations of the three different $m_J$ states (summing over $m_I$ states) which reach the surface as a function of the applied magnetic field integral in the first electromagnet, $B1$. The projections are defined using the surface normal as the quantisation axis. **d** Schematic classical depiction of the $m_J = 1$ and $m_J = -1$ 'helicopter' states and the $m_J = 0$ 'cartwheel' state.

It is at this point, where the approach we use deviates from conventional deflection experiments. The beam, which is now no longer equally populated in all nine states, exits the hexapole field adiabatically into a dipole field along the $Z$ direction (Dip1). The strength of the magnetic field gradients within the hexapole lens (>T mm$^{-1}$) results in initially pure $m_I$, $m_J$ states in the Z frame of reference[32]. The beam then enters non-adiabatically (i.e., the direction of the magnetic dipole does not follow the change in the direction of the local magnetic field) into the first solenoid (Solenoid1) which generates an electromagnetic field, $B1$, oriented along the -$X$ axis (antiparallel to the beam axis). As the molecules were previously in pure $m_I$, $m_J$ states in the Z frame of reference, they are in superpositions of the nine $m_I$, $m_J$ states defined along the quantisation axis of the $B1$ field. All nine complex amplitudes continuously change as the beam propagates through the beam line until the beam reaches the surface located in the scattering chamber. The coherent evolution of these amplitudes is given by both the field dependent and the field independent terms of the Ramsey Hamiltonian[25] given by Supplementary Eq. (2). Thus, if both the velocity of the particle and the magnetic field profiles through the instrument are known, we can calculate the evolution of the quantum states exactly and coherently control the states that reach the sample.

Figure 2b shows an example for a calculation of the changes in the nine $m_I$, $m_J$ states that reach the sample as a function of the strength of $B1$ where the quantisation axis is taken as the surface normal, $Z_N$, Fig. 2c compares the populations of the three $m_J$ rotational states that reach the sample and Fig. 2d shows the classical view of the different $m_J$ states. For simplicity we only plot the square of the amplitudes of these states in Fig. 2b, c. However, as the control is coherent, we also know the relative phases of these superposition states. Figure 2b, c illustrates our ability to have more molecules in a particular $m_I$, $m_J$ state reach the surface by choosing a particular magnetic field ($B1$) value.

Once the beam approaches the surface it can scatter into one of the diffraction channels and the quantum state changes again, this time due to the interaction potential with the surface. This change, which reflects the physics and chemistry of the collision, can be described using the scattering matrix, (S-matrix), which relates the molecular wave function before and after scattering[22]. Obtaining the S-matrix, and the corresponding insight into the molecule–surface interaction potential, is the goal of our experiment. The surface is mounted on a six-axis manipulator, which allows the scattering angle to be changed, allowing us to perform measurements of different diffraction channels, each of

which is characterised by a different evolution of the quantum states and correspondingly a different S-matrix.

After scattering, a certain fraction of the molecular beam that corresponds to a particular diffraction channel travels through the second arm of the instrument. Analysing the total flux of the scattered beam, while modulating the $B1$ field already provides information about the sensitivity of scattering to the incoming rotational state, and allows qualitative comparisons between the stereodynamic response of different types of surfaces[24]. However, in order to perform a quantitative state-to-state experiment, we need further magnetic manipulation combined with a detailed interpretation scheme. The magnetic manipulation includes a second electromagnet (Solenoid2) with a magnetic field strength of $B2$ directed along the $-X'$ axis. In this second electromagnet, the scattered wave function again evolves coherently; this evolution can be controlled by changing the field strength, $B2$. The molecules then pass through a second dipole (Dip2) before entering a second hexapole field[33] (Hex2), which transmits them towards a particle detector[34], with probabilities which depend on their magnetic moment projection along the $-Z'$ axis.

**Determining the scattering matrix.** The circle markers in Fig. 3 are the intensity of a diffracted $H_2$ beam as a function of the magnetic field-integral values in the first solenoid ($B1$). The measurements performed on the (1,0) and (−1,0) diffraction peaks are plotted in Fig. 3a, b, respectively. These are the average of at least five identical $B1$ scans, with the error bars reflecting the uncertainty in the data estimated from the scatter of the measured values. Two striking features of both datasets are the relatively strong amplitude of the oscillations in the measured intensity as a function of the magnetic field value, and the fact that the oscillations continue without decaying within the full range of the measurement, producing a rich and rather complex pattern. The

large oscillation amplitudes reflect a large dependence of the scattering probabilities on the rotational orientation of the $H_2$ molecules. The complex pattern of the signal is related to the Rabi-oscillations within a nine-level quantum system[24]. Our ability to coherently control these oscillations for a relatively large range of magnetic fields is related to the high angular resolution of the apparatus, which translates into a very narrow range of beam energies when measuring a diffraction peak. Similar experiments can be performed for specular scattering, but the wider velocity distribution of the molecules that contribute to the signal leads to a faster decay of the oscillation amplitude.

As the Hamiltonian, the velocity, and the static magnetic and electromagnetic field profiles of our apparatus are known quantities, the signals shown in Fig. 3 (related to the square of the wave function given in Supplementary Eq. (3)) can be related to one unknown property, the scattering matrix. This S-matrix, which can be defined as the operator relating the molecular wave function just before scattering to that immediately after scattering[22], is a property that expresses the effect of the molecule–surface interaction potential as it approaches, collides and moves further away from the surface. Previously, this descriptor was only accessible in theoretical calculations, and the accuracy to which it could be determined depends on the accuracy of these calculations. Experiments, on the other hand, could only access properties related to the sum of the square of the S-matrix elements, such as the total scattering intensity into a particular diffraction channel[35–39], or the degree of rotational alignment[40–44]. Since the observable measured in our experiment depends on the coherent manipulation of the wave function both before and after scattering, the S-matrix itself can now be determined from the experiment as we show below.

Even for the simple case of ortho-$H_2$ scattering without exchanging energy with the surface and without changing its overall rotational state ($J = 1$), which is the predominant contribution in the experiments we present, the corresponding $9 \times 9$ S-matrix consists of 81 complex elements, where the magnitude of the elements corresponds to the state-to-state scattering probabilities. Fortunately, the mixing of the $m_I$, $m_J$ states is completely negligible within the short time scale of the molecule–surface interaction (pico-seconds). Combining this short mixing time, with the reasonable assumption that the nuclear spin does not affect or take part in the collision (LiF is a non-magnetic surface), we can completely describe the collision using the much simpler $3 \times 3$ S-matrix of the $m_J$ subspace, which is expanded to relate to the $9 \times 1$ states before and after scattering. It is important to note that we chose the quantisation axis to be the surface normal. This choice is arbitrary and does not affect the measured quantities, but follows the common practice used in theoretical gas–surface scattering[22].

A further important simplification can be made to the scattering matrix used to fit the data due to the reflection symmetry of the LiF(100) surface, and consequently the molecule–surface potential. This symmetry means that we do not expect molecules rotating as clockwise helicopters in the surface plane to scatter with different probabilities to anti-clockwise helicopters, allowing us to constrain the scattering matrix to have identical magnitudes for the $m_J = 1$ and $m_J = -1$ elements.

The large number of experimental measurements performed at different $B1$ and $B2$ values for each diffraction peak allows us to extract an S-matrix by fitting the entire dataset. The dashed lines in Fig. 3 show the simulated signal for the S-matrix that produces the best fit to the experiment obtained using the procedures described in the methods section and in more detail in the Supplementary notes 1 and 2. Table 1 and Supplementary Table 1 show the values of the square of the amplitudes of the S-matrix

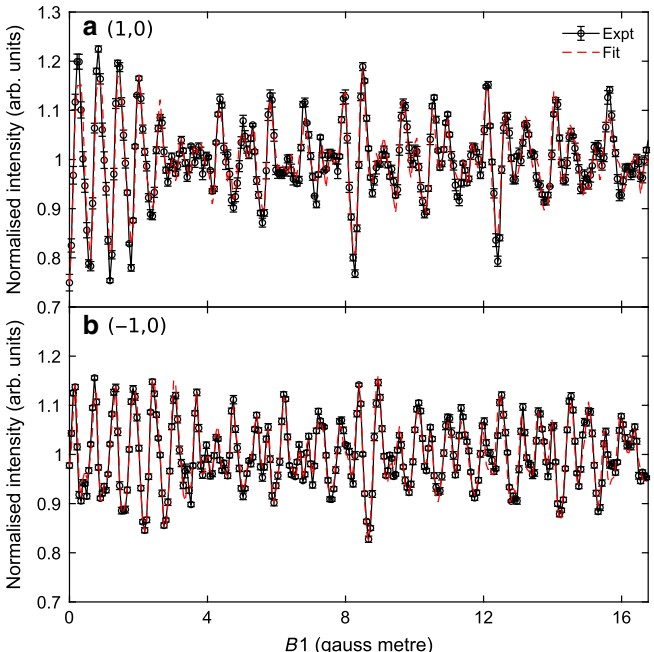

**Fig. 3 Experimental data and fits. a** The normalised intensity of $H_2$ scattered from LiF(100) into the (1,0) diffraction channel as a function of the applied field integral in the first electromagnet, $B1$, for $B2 = 0$ G m (black circles) and the fit to the data (red dashed lines). The error bars represent standard errors from repeated $B1$ scans. **b** As for panel a, but for $H_2$ scattering into the (−1,0) diffraction channel.

**Table 1 Relative state-to-state scattering probabilities.**

| Elements | Initial $m_J$ | Final $m_J$ | (1,0) peak | (−1,0) peak |
|---|---|---|---|---|
| a | 1 | 1 | 0.2 | 0.04 |
| b | 0 | 1 | 0.6 | 0.4 |
| c | −1 | 1 | 1.0 | 1.0 |
| d | 1 | 0 | 0.1 | 0.4 |
| e | 0 | 0 | 0.2 | 0.05 |
| f | −1 | 0 | 0.1 | 0.4 |
| g | 1 | −1 | 1.0 | 1.0 |
| h | 0 | −1 | 0.6 | 0.4 |
| i | −1 | −1 | 0.2 | 0.04 |

The values of the relative rotational orientation ($m_J$) state-to-state scattering probabilities for $H_2$ scattering from LiF(100) into the (1,0) and (−1,0) diffraction peaks. Note the S-matrix is not unitary as it provides the probabilities for a particular diffraction channel. For convenience, we normalised the elements to the 1–1 element. The probabilities presented here have been calculated using the amplitudes obtained from the fit, as opposed to the rounded values presented in Supplementary Table 1. The estimated uncertainty in the relative probabilities is 20%. For a discussion of the uncertainty in these values, see Supplementary note 4.

elements which provide relative state-to-state scattering probabilities and the S-matrix elements for the two diffraction peaks respectively. The simultaneous fit to a large number of experimental data points, results in a well-converged result for the S-matrix elements. A detailed description of the uniqueness of the extracted values is presented in Supplementary note 3.

Previous work on $H_2$ scattering from LiF has suggested a $\Delta m_J = 0$ propensity rule[45,46], i.e., a collision with a LiF surface cannot change the rotational plane of a $H_2$ molecule. The results presented here show a breakdown of this propensity rule, with the off-diagonal elements of the scattering matrix (corresponding to $\Delta m_J \neq 0$) being on the same order of magnitude as the diagonal elements ($\Delta m_J = 0$). This supports a previous theoretical prediction made for this system using simplistic models which account for the interaction between the electrical quadrupole of $H_2$ and the surface ions[23], and the use of $\Delta m_J \neq 0$ collisions in the interpretation of Knudsen flow experiments for $H_2$ on LiF[47,48]. Recent calculations have shown that these rotational flip transitions, which we can now determine directly from experimental measurements, are closely linked to reactive adsorption events and particularly relevant for an atomistic understanding of heterogeneous catalysis[49].

**Stereodynamic effects.** By taking the square modulus of the elements from our empirical scattering matrix, i.e. extracting the $m_J$ state-to-state scattering probabilities, we can quantitatively assess another long-standing theoretical prediction made for the $H_2$–LiF system, which is that the collisions will rotationally polarise the scattered $H_2$ beam[23]. This corresponds to the populations in the $m_J = 1$ and $m_J = −1$ (helicopter) states after the collision being different to the $m_J = 0$ (cartwheel) state, a phenomenon often referred to as surface stereodynamics[40,50]. As we have access to all the state-to-state probabilities, we can look at partial summations of these elements. For example, comparing the sum of the rows a–c in Table 1, which corresponds to the relative population in $m_J = 1$ after the collision, and the sum of rows d, e and f (relative population in $m_J = 0$ after the collision) shows that both diffraction channels polarise the rotational orientation of $H_2$. In both cases, more molecules are rotating like helicopters after the collision than cartwheels, confirming the prediction that LiF can be used to polarise $H_2$ rotations.

Earlier studies have shown that rotationally polarised hydrogen ($D_2$) molecules can be obtained from the recombinative desorption of D atoms from a Cu(111) surface, where the molecules were found to preferentially desorb rotating like

helicopters rather than cartwheels[43]. Collisions of gas phase molecules with surfaces have also previously been demonstrated to create rotational polarisation in scattered molecules, with an Ag(111) surface shown to create rotational alignment in scattered NO[44], and both rotational alignment and orientation in scattered $N_2$[42,43]. In both cases, strong negative alignments were measured in rotationally inelastic scattering. This corresponds to collisions that change the rotational angular momentum, $J$, but tend to conserve $m_J$. In contrast, the rotational alignment that is created for $H_2$ scattering from LiF in the present study arises within rotationally elastic scattering ($\Delta J = 0$), and is due to $m_J$ changing collisions. The observation of differences in the scattering probabilities of different rotational orientations can be qualitatively explained in terms of the different potential energy surfaces seen by the different $m_J$ state molecules, with molecules in $m_J = 1$ and $m_J = −1$ experiencing a more corrugated potential than the molecules in $m_J = 0$[23]. In contrast, obtaining quantitative predictions of the relative populations within a scattered beam, such as calculating whether we expect an increased helicopter/cartwheel population in a particular channel, requires calculating the constructive interference of the different wave functions within that diffraction channel. Consequently it seems simpler explanations based on a classical picture of the collision are unlikely to be helpful and a quantum mechanical analysis of the interaction is needed.

The initial rotational orientation of molecules has also been shown to change how molecules interact with a surface in previous studies which have used collision induced rotational polarisation in molecular beam expansions[51], paramagnetic[30] and vibrationally excited[52] molecules to prepare molecules with an anisotropic distribution of $m_J$ states before the gas–surface collision. The experimental method we employ, which does not perturb the molecular ground state, allows us to also study how the quantum state (rotational orientation) of the $H_2$ molecule just before the collision, changes the probability of the molecule to scatter into a particular diffraction channel (regardless of its final quantum state). For the (1,0) diffraction peak, the relative scattering probabilities of $H_2$ molecules initially in $m_J = 1$ (which can be found by adding rows a, d and g in Table 1) is less than for $H_2$ molecules which were in $m_J = 0$ (which is found by adding rows b, e and h), showing that molecules that are rotating like helicopters are less likely to scatter into the (1,0) diffraction channel than molecules rotating like cartwheels. The reverse is true for the (−1,0) diffraction channel, where molecules that are rotating like helicopters in $m_J = 1$ or $m_J = −1$ are more likely to scatter into that channel than molecules rotating like cartwheels. Consequently, the intensity of the two diffraction channels not only depends on the $J$ state populations as reported previously[53] but also depends on the initial $m_J$ state populations of the hydrogen in the molecular beam. This suggests that information about the rotational orientation of a $H_2$ molecular beam could be obtained by comparing the intensities of the diffraction peaks for $H_2$ scattering from a LiF crystal, i.e., the crystal can potentially also be used as a rotational orientation analyser.

**Summary.** We have demonstrated the use of a coherent magnetic field control technique to obtain a complete state-to-state stereodynamics analysis of $H_2$ colliding with a LiF(100) surface. The magnetic coherent control of the molecule, applied both before and after the scattering event, allows us to measure the evolution of molecular quantum wave functions during the collision, expressed by the nine complex-valued elements of the $m_J$ scattering matrix. Thus the measurements provide unique experimental access to a fundamental descriptor of the molecule–surface interaction.

The relative state-to-state scattering probabilities have shown that collisions which change the direction of the rotational plane of $H_2$ ($\Delta m_J \neq 0$) are significant, confirming a theoretical prediction[23] that has also been beyond the reach of other existing state-of-the-art surface-science experiments. Our results which simultaneously quantify the stereodynamic effects both before and after the collision, introduce a stringent type of characterisation for molecule–surface dynamics, and supply the data needed for using a LiF surface as a rotational orientation polariser and analyser.

A particularly exciting opportunity made possible by experimentally determined S-matrices is related to the development of theoretical models for molecule–surface interactions. Significant efforts are being made to develop reliable multi-dimensional potential energy surfaces, which can be used to study molecule–surface collisions and heterogeneous catalysis[54–56]. Up to now, the probabilities that molecules scatter into elastic and inelastic diffraction channels provided a sensitive way of benchmarking theoretical interaction models[35–38,57–59], in addition to state-resolved sticking measurements which provide valuable complementary information for the reaction probabilties[60–62]. Comparing calculated S-matrices, once these become available, with experimentally determined values of the type reported in this paper, will provide an extremely sensitive, and particularly valuable benchmark for assessing theoretical models.

Finally, we note that the coherent manipulation experiments and the analysis methods presented in this work rely on the rather general phenomena of the rotational magnetic moment, and are not restricted to $H_2$ molecules. As such the technique could be used to study both rotationally elastic and inelastic scattering of ground state molecules including HD, $H_2O$, $NH_3$, $CH_4$ and other small molecules from various metals and insulators, allowing us to obtain empirical scattering matrix benchmarks for a range of systems.

## Methods

**Experimental methods**. The apparatus used in the present study is shown schematically in Fig. 1[24]. The supersonic molecular beam was formed by expanding pressurised hydrogen (research grade) through a 30 μm diameter nozzle cooled to 100 K. The average kinetic energy of the beam, which was determined from the known lattice vector of the LiF surface and the angular position of the diffraction peaks was 22 meV.

The sample was prepared by cleaving (in air) a single crystal lithium fluoride sample (Crystran Ltd) and transferring it within minutes into an ultra-high vacuum (UHV) chamber ($P = 10^{-10}$ mbar), where it was mounted on a home-built non-magnetic six-axis sample manipulator with heating, cooling and sample transfer capabilities. The crystal was flash annealed to 450 K and the quality of the surface was verified by obtaining a very narrow specular peak (FWHM 0.07°) and the expected diffraction pattern.

All the measurements presented in this paper were performed at a surface temperature of 165 K. At these temperatures the sample remained inert and no degradation of the specular signal was seen within the measurement time. Other experiments performed at lower temperatures (135 K) showed an essentially identical oscillation curve, however, at these lower temperatures the signal intensity degraded slightly after long periods (>3 h), likely due to adsorption of water molecules. The crystallographic azimuths were determined using the known diffraction pattern of the (100) surface, with an estimated uncertainty < 0.5°.

The magnetic fields, $B1$ and $B2$, were created by passing currents through two high-homogeneity solenoids. The currents were scanned using two independent high-stability power supplies (Danfysik) calibrated to control currents on a ppm level over a 0–10 A range. The solenoids are enclosed in a triple layer mu-metal magnetic shield to protect from stray magnetic fields. The UHV sample chamber is constructed from mu-metal and includes an additional internal mu-metal cylinder to further reduce residual fields penetrating into the region where the molecules travel. The three-axis magnetic field profiles of the beam line were measured by inserting a sensitive gauss meter (AlphaLab Vector Gauss Meter) and scanning it along all the regions in the beam line where the wave functions evolve coherently (essentially from the first dipole field Dip1 to the second dipole field Dip2). Details of the first and second hexapole have been published previously[26,33].

**Data analysis methods**. Interpreting the experimental data and extracting an S-matrix is achieved by combining a detailed simulation of the evolution of the magnetic molecular states through the apparatus with an error-minimising fitting algorithm applied to a large number of experiments with different magnetic field values. Both of these procedures, which are described in detail in the Supplementary notes 1 and 2 and Supplementary Figs. 1 and 2, are outlined briefly below.

Two types of calculations are used to simulate the propagation of the molecular waves through the apparatus. Within the two hexapole fields (Hex1 and Hex2), which are characterised by large magnetic field gradients and correspondingly pure $m_I$, $m_J$ states[32], semi-classical ray tracing calculations are used to determine the particle trajectories and corresponding transmission probabilities for each state[63]. For the majority of the beam line, contained between these two hexapole fields, the propagation of the wave function needs to be calculated coherently. This second type of calculation involves solving the magnetic Ramsey Hamiltonian[25] quantum mechanically while propagating the molecular centre of mass classically[24] which has been shown to be essentially identical to a fully quantum calculation for static scattering events[64]. The coherent evolution of the wave functions is calculated through the 3-d magnetic field profiles of the beam line for each of the $B1$ and $B2$ values used experimentally.

If the propagation is accounted for accurately, the only unknown factor contributing to the magnetic field dependent signal intensity, is the S-matrix (see Supplementary Eq. (5)). Starting with random S-matrix values, the downhill simplex method of Nelder and Mead[65] is combined with a simulated annealing algorithm to minimise the difference between the simulated signal and the experimental data. A key point, is that the algorithm searches for a simultaneous best fit to a large number of experimental points (602 $B1$, $B2$ pairs, achieved by scanning 301 $B1$ values for two different $B2$ values). This minimisation procedure is repeated 150 times with randomised initial parameters to ensure the result of the fit gives a converged scattering matrix that corresponds to the global minimum of the fit. Further details of the fitting procedure, the tests that were done to ensure the results correspond to a unique, converged S-matrix and the uncertainties related to this procedure can be found in Supplementary Notes 3 and 4 and Supplementary Figs. 3–8.

**Reporting summary**. Further information on research design is available in the Nature Research Reporting Summary linked to this article.

## Data availability
The data that support the findings of this study are available from the corresponding author upon request.

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

## Acknowledgements

The authors would like to thank Dan Auerbach for critically reading an early version of this paper, and thank Bill Allison, Geert-Jan Kroes, Dick Manson, Greg Sitz and Mark Somers for stimulating scientific discussions. The authors are also grateful to Nadav Avidor and David Ward for their kind assistance with setting up the experimental system. We acknowledge the support of the Supercomputing Wales project, which is part-funded by the European Regional Development Fund (ERDF) via the Welsh Government. This work was funded by an ERC consolidator grant (Horizon 2020 Research and Innovation Programme grant 772228).

## Author contributions

G.A. conceived and supervised the project. Y.A. performed the measurements. H.C. performed the analysis of the measurements. H.C., O.G., J.T.C. and G.A. developed the analysis methodology. T.M. developed the wave function propagation algorithm. O.G., H.C. and I.L. developed different aspects of the analysis code. O.G., Y.A., M.B. and H.L. installed, improved and characterised the experimental setup. G.A. and H.C. wrote the paper.

## Competing interests

The authors declare no competing interests.
