## [Peer Review File · Nature Communications]

This is a very exciting study. The authors design a novel molecular beam apparatus that is able to reveal the unprecedentedly detailed dynamics for molecule-surface scattering, namely the quantum state-to-state scattering matrix. Commonly used magnetic field based techniques only enables the magnetic state splitting and deflection in the magnetic field, but the couplings between nuclear and rotational magnetic states would mix the resultant m_i and m_j states into a superposition within micro-seconds after flying away the magnetic field. However, after the molecules pass through the magnetic field, here these authors further propagate the beam through another electromagnetic field and allows coherent evolution of all nuclear-spin and magnetic fine states in superposition status, until the beam approaches the surface. By doing the reverse procedure before detecting the molecule scattered from the surface, state-to-state S-matrix can be extracted for all involved magnetic states by adjusting the S-matrix elements to fit experimental signals. This is a unique technique that combines the exquisite experimental design with numerical wavepacket simulations. They use this method to study H_2 diffraction from a LiF surface and reveal the high probability of magnetic state flipping and the possibility of polarize the rotational orientation of H_2 , in good agreement with earlier theoretical predictions. This work really opens new opportunity for understanding molecule-surface interaction at quantum-state level and offers new benchmark guiding the development of theoretical models for molecule-surface systems. Given these achievements, I think this work meets the high standard of research articles required by Nat. Commun. and has the potential to attract general readership of Nat. Commun. I believe the new concept that the authors have introduced to accomplish this experiment will stimulate many future studies in this field and others. Having said that, I do have several questions that I would like the authors to address, as listed below.

1. The experiment presented in this work is a bit different from conventional molecular beam experiments. If I understand correctly, the s-matrix is not directly measurable but indirectly extracted, starting from a random guess, followed by optimizing the elements so that the probabilities of final states (square of the product of S-matrix and other transformation matrices) fit the signals from the detector. In other words, the accuracy of the method is highly dependent on this non-linear optimization. In conventional methods, the experimental quantity should have an uncertainty simply estimated by repeated measurements in the same conditions, which corresponds to systematic errors. In the present work, there should be also the error coming from the optimization itself. These errors are not accumulated linearly. How do the authors properly account for their given uncertainties, e.g. 20% in Table 1. In addition, the calculated populations of different m_J states seem highly oscillating in Fig. 2c. This suggests that the measured S-matrix should be very sensitive to the length and field strength of B1. Could this be part of error?
2. This work observes the non-conservative magnetic transition during H_2 scattering from LiF, suggesting the possibility of using a LiF surface to polarize the H_2 rotational orientation and confirming previous theoretical predictions. Also, it is found that the H_2 molecules, with $m_J=0$ more likely scatter to the (1,0) diffraction

channel while these with the $m_J=1/-1$ prefer to scatter into the $(-1,0)$ diffraction channel. But there seems no simple physical picture for this observation, unlike that often present in many other joint experimental-theoretical works. Although this is partially understandable because these new benchmark data provide stringent challenges for new theory, it is better to offer some illustrative or graphic interpretations, e.g. any relation to the anisotropy or topography of potential energy surface in the entrance channel or corrugation of the surface? Or is it largely related to the overlaps between initial and final state wavefunctions? In addition, the authors should compare their results with theoretical predictions more quantitatively so that any disagreement may guide the further development of potential energy surface and/or quantum scattering theory. More discussions on these issues are necessary.

3. As I have said, this work presents an impressive and unique technique for measuring state-to-state scattering quantities between rotationally elastic magnetic states. But it is not clear that whether this experimental method can be used to study rotational or vibrational inelastic scattering of molecules on solid surfaces. Its limitations, if there were, should be discussed shortly.

I conclude my comments with an optional suggestion,

Following my third question, recent state-to-state scattering experimental and theoretical advances for polyatomic molecules should be referenced, especially if the current method can be extended to study these systems. For example, the Beck group reported vibrationally inelastic state-to-state scattering experiments of CH_4 on $\text{Ni}(111)$ [see PRL, 120, 053402] and the Jiang group presented state-to-state quantum scattering calculations for H_2O on $\text{Cu}(111)$ [see PRL, 123 106001].

Reviewer #2 (Remarks to the Author):

The interaction of hydrogen molecules with solid surfaces has been a model system for understanding gas-surface interactions. Over the years, the experimental measurements and theoretical understanding of this process have become increasingly more sophisticated. The experiments described in this paper take the experimental measurements to the next level. This is quite well summed up in the first few sentences of their abstract: "The coherent evolution of a molecular quantum state during a molecule-surface collision is a detailed descriptor of the surface interaction potential which was so far inaccessible to measurements. We used a magnetically controlled molecular beam technique to study the collision of rotationally oriented ground state hydrogen molecules with a lithium fluoride (LiF) surface. The coherent quantum control nature of the technique we developed allowed us to measure the changes in the complex amplitudes of all the rotational projection quantum states, and express them using a scattering matrix formalism."

This is a clearly written and coherent exposition on the new methods being innovated in this research effort. It has many salient points as noted above. I wish to emphasize that the ability to probe experimentally the evolution of molecular wave functions is quite novel, and leads in a straightforward manner to determination of the scattering S-matrix. This is important as it provides a benchmark for theoretical analyses of hydrogen-surface interactions, providing a rigorous test of both interaction potentials and scattering methodologies.

The only point that needs some moderation is the claim that this work is the first to discuss using gas-surface collisions to prepare rotationally polarized or oriented species. This has been discussed previously by several groups in concept based upon molecule-surface scattering experiments that are rotationally analyzed with respect to final quantum state, both for paradigmatic molecular hydrogen interactions in angle-resolved state-to-state scattering experiments as well as laser interrogated quantum-resolved scattering.

Some key literature citations are clearly missing and need to be added in the proper places and referenced accordingly. Early, seminal work especially for the molecular hydrogen/Ag(111) system (3 references below) must be included as that system provided the best benchmarks based upon combined scattering experiments and extensive quantum scattering calculations encompassing elastic, inelastic, and resonant scattering including some aspects of magnetic sub-level effects. A key paper on hydrogen-graphite also given below, and also needs to be cited. Other papers on scattering from alkali halides could also be added as deeper background if desired.

"Investigation of the Spatially Anisotropic Component of the Laterally Averaged H₂/Ag(111) Physisorption Potential" K.B. Whaley, C.-F. Yu, C.S. Hogg, J.C. Light, and S.J. Sibener
J. Chem. Phys. 83, 4235-4255 (1985)

"Investigation of the Spatially Isotropic Component of the Laterally Averaged H₂/Ag(111) Physisorption Potential" C.-F. Yu, K.B. Whaley, C.S. Hogg, and S.J. Sibener
J. Chem. Phys. 83, 4217-4234 (1985).

"Selective Adsorption Resonances in the Scattering of n-H₂, p-H₂, n-D₂, and o-D₂ from Ag(111)"
C.-F. Yu, K.B. Whaley, C.S. Hogg, and S.J. Sibener
Phys. Rev. Lett. 51, 2210-2213 (1983).

"Selective adsorption of 1H₂ and 2H₂ on the (0001) graphite surface" L. Mattera, F. Rosatelli, C. Salvo, F. Tommasini, U. Valbusa Surf. Sci. 93, 515-525 (1980)

A final comment pertains to the last paragraph of the paper. The authors briefly allude to the generality of their new methods: "Consequently, there is significant potential for using these methods to obtain empirical scattering matrix benchmarks and explore state-to-state stereodynamic effects for a range of ground state molecules colliding with various different surfaces, as well as in gas phase scattering." I think the readership would appreciate some elaboration on this point: What are some other illustrative ground state molecules that these methods can be realistically applied to, what surfaces, and which gas phase systems? What questions can be addressed if these next directions are pursued? Doing this will nicely put into context that these approaches are indeed more general than the test case demonstrated for molecular hydrogen/LiF given herein.

In summary, this is an excellent and in several aspects pioneering paper. My clear recommendation is to proceed with publication in Nature Communications after the aforementioned straightforward changes and necessary additions are enacted. Well done.

Review of manuscript: NCOMMS-20-13736-T

Manuscript title: "Coherent control of rotational orientation states: a new ultra-sensitive benchmark for modelling gas-surface interactions"

Authors: Y. Alkoby et al.

This manuscript reports an experimental study of the scattering of rotationally oriented ground state H₂ molecules from a LiF(100) surface. By using a coherent magnetic field control technique, the authors were able to follow the evolution of molecular quantum wave functions during the collision, which allows performing a complete state-to-state stereodynamics analysis of the H₂ scattering process. A quite remarkable result is the first experimental determination of a scattering matrix, which will provide unique experimental access to a fundamental descriptor of the molecule-surface interaction. Special attention has been paid to analyzing the accuracy and reproducibility of the S-matrix determination, presented in the Supplementary Information. Moreover, these results show that both diffraction channels polarise the rotational orientation of H₂, making it possible the use of LiF(100) as a rotational orientation polarizer.

This is a significant and novel work that will extend the applications of this unique technique to the broad field of stereodynamics. The study has been conducted by one of the leading groups in the field, and will certainly be of great interest to the journal's readership. The article is written clearly and concisely. Subject to what I discuss below, I recommend publication of a suitably revised version of the manuscript in *Nature Communications*.

- When the current results are compared with previous calculations for the same system performed by Kroes et al., the authors should also mention the first experimental confirmation of large differences in the diffraction of normal- and para-H₂ reported by Bertino and coworkers, Phys. Rev. Lett. 81 (1998) 5608.
- Although the observed rotational flip transitions are already present in frozen surface calculations (like the ones reported in Ref. 23), one can expect that surface phonons may influence the flip-mechanism. The authors mention in the Supplementary Information that essentially the same oscillation curves are observed at 165 K and 135 K. I wonder if there is some evidence for a different behavior at higher surface temperatures?
- The differences observed for the (1,0) and (-1,0) diffraction peaks (Fig.3a,b) are quite clear, in part due to the small error bars. I understand that the small error bars correspond to the small uncertainty when measuring the diffraction intensities. I think this information is relevant and should be added to the main text.
- Second order diffraction is also observed from this surface (see paper by Bertino et al. mentioned above). I guess the current work is focused on the first order diffraction peaks, (1,0) and (-1,0), due to their higher intensity. However, it would be interesting to discuss what kind of additional information could be obtained by performing a similar analysis to second order diffraction peaks, which in principle should be possible using the authors' experimental setup.

The comments from the Reviewers are reproduced in black below, with our replies written in blue. Any changes that have been made to the manuscript are given both here and in the revised manuscript in red. We thank the Reviewers for their helpful comments, and address each of them in turn below.

Reviewer 1

This is a very exciting study. The authors design a novel molecular beam apparatus that is able to reveal the unprecedentedly detailed dynamics for molecule-surface scattering, namely the quantum state-to-state scattering matrix. Commonly used magnetic field based techniques only enables the magnetic state splitting and deflection in the magnetic field, but the couplings between nuclear and rotational magnetic states would mix the resultant m_I and m_J states into a superposition within micro-seconds after flying away the magnetic field. However, after the molecules pass through the magnetic field, here these authors further propagate the beam through another electromagnetic field and allows coherent evolution of all nuclear-spin and magnetic fine states in superposition status, until the beam approaches the surface. By doing the reverse procedure before detecting the molecule scattered from the surface, state-to-state S-matrix can be extracted for all involved magnetic states by adjusting the S-matrix elements to fit experimental signals. This is a unique technique that combines the exquisite experimental design with numerical wavepacket simulations. They use this method to study H₂ diffraction from a LiF surface and reveal the high probability of magnetic state flipping and the possibility of polarize the rotational orientation of H₂, in good agreement with earlier theoretical predictions. This work really opens new opportunity for understanding molecule-surface interaction at quantum-state level and offers new benchmark guiding the development of theoretical models for molecule surface systems. Given these achievements, I think this work meets the high standard of research articles required by Nat. Commun. and has the potential to attract general readership of Nat. Commun. I believe the new concept that the authors have introduced to accomplish this experiment will stimulate many future studies in this field and others.

We thank the Reviewer for their extremely positive comments.

Having said that, I do have several questions that I would like the authors to address, as listed below.

1. The experiment presented in this work is a bit different from conventional molecular beam experiments. If I understand correctly, the s-matrix is not directly measurable but indirectly extracted, starting from a random guess, followed by optimizing the elements so that the probabilities of final states (square of the product of S-matrix and other transformation matrices) fit the signals from the detector. In other words, the accuracy of the method is highly dependent on this non-linear optimization. In conventional methods, the experimental quantity should have an uncertainty simply estimated by repeated measurements in the same conditions, which corresponds to systematic errors. In the present work, there should be also the error coming from the optimization itself. These errors are not accumulated linearly. How do the authors properly account for their given uncertainties, e.g. 20% in Table 1.

A description of the uncertainties which arise from the fitting procedure are described in Section S4 of the Supplementary Information. Here we show that fitting simulated data with noise levels that are comparable to those in the experimental data returns the S-matrix amplitudes that were used to simulate the data within 10% (Figure S7). As the probabilities presented in Table 1 are the square of these values, this leads to an approximate uncertainty

of 20% in these values. We have made minor changes to the text to further emphasise that this information is given in the SI by rephrasing the following text on page 7.

Further details of the fitting procedure, the tests that were done to ensure the results correspond to a unique, converged S-matrix and the uncertainties related to this procedure can be found in the SI.

In addition, the calculated populations of different mJ states seem highly oscillating in Fig. 2c. This suggests that the measured S-matrix should be very sensitive to the length and field strength of B1. Could this be part of error?

The Reviewer is correct that any errors in the magnetic field profiles can introduce errors into the S-matrix elements that are obtained through the fitting procedure. However, as detailed in Section S4 of the Supplementary Information, fits have also been done where the length of the magnetic field profile and field strength of B1 have been changed to determine what effect this has on the S-matrix we obtain from the experimental data. We find that the S-matrix elements change by a few percent, but by less than our stated 10% uncertainty in the values.

2. This work observes the non-conservative magnetic transition during H2 scattering from LiF, suggesting the possibility of using a LiF surface to polarize the H2 rotational orientation and confirming previous theoretical predictions. Also, it is found that the H2 molecules with $mJ=0$ more likely scatter to the (1,0) diffraction channel while these with the $mJ=1/-1$ prefer to scatter into the (-1,0) diffraction channel. But there seems no simple physical picture for this observation, unlike that often present in many other joint experimental-theoretical works. Although this is partially understandable because these new benchmark data provide stringent challenges for new theory, it is better to offer some illustrative or graphic interpretations, e.g. any relation to the anisotropy or topography of potential energy surface in the entrance channel or corrugation of the surface? Or is it largely related to the overlaps between initial and final state wavefunctions? In addition, the authors should compare their results with theoretical predictions more quantitatively so that any disagreement may guide the further development of potential energy surface and/or quantum scattering theory. More discussions on these issues are necessary.

We would like to emphasise that while we use theory to interpret the propagation of the molecular quantum states through the apparatus, the S-matrix values we obtain are determined from the experimental measurements rather than from theory. Indeed it would be very valuable to compare with theory, whether the results agree or not. Unfortunately, a theoretical description of this particular system which produces a theory-based S matrix which we can compare with is still not available in the literature (we are aware of one ongoing attempt to produce DFT based scattering matrices for this system, but this has still not materialised to a result we can compare with).

Direct comparison with the theoretical model of reference 23 is not possible due to the differences in the beam energies, geometry and rotational state. Furthermore, even if calculations of the type described in reference 23 were redone for the relevant experimental conditions, a more modern theoretical approach, which tries to mimic the molecule-surface interaction as accurately as possible, would probably be more suitable for such a comparison. We hope that this work will stimulate interest in the theoretical chemistry community, and will result in calculations being done which are quantitatively comparable with the

experiments that have been done here, which can then be used as the Reviewer states to guide the development of accurate potential energy surfaces and quantum scattering theories.

Regarding an illustrative/graphical interpretation, as we state in the last paragraph of page 11 and first paragraph of page 12, qualitative descriptions can and have been used to explain the existence of differences between the scattering probabilities of different m_j states, linking them to the corrugation of the potential. However, the actual scattering matrix values, and the corresponding trends in terms of which rotational projection states are enhanced in a particular scattering channel, reflect the coherent interference of the quantum states. As a result a detailed and accurate theory is needed to understand the experimental data, and heuristic/graphical explanations are not likely to be meaningful.

We have made minor changes on page 11 and 14 to clarify the text.

Page 11:

The observation of differences in the scattering probabilities of different rotational orientations can be qualitatively explained in terms of the different potential energy surfaces seen by the different m_j state molecules, with molecules in $m_j = 1$ and $m_j = -1$ experiencing a more corrugated potential than the molecules in $m_j = 0$ ²³.

Pages 13 to 14:

Comparing calculated S-matrices, once these become available, with experimentally determined values of the type reported in this manuscript, will provide a new, extremely sensitive, and particularly valuable benchmark for assessing theoretical models.

3. As I have said, this work presents an impressive and unique technique for measuring state-to-state scattering quantities between rotationally elastic magnetic states. But it is not clear that whether this experimental method can be used to study rotational or vibrational inelastic scattering of molecules on solid surfaces. Its limitations, if there were, should be discussed shortly.

For H_2 at the collision energy of the experiment only rotationally elastic ($\Delta J=0$) scattering will be observed. However, if other molecules such as HD, NH_3 , H_2O and CH_4 are used it should be possible to study rotationally inelastic scattering of molecules with surfaces.

With the current experimental set-up, it would not be possible to study vibrational inelastic scattering from a vibrationally excited state to the ground vibrational state. However, other state-of-the-art techniques exist which can be used to study rotational orientation effects in vibrationally excited molecules using photo-excitation. The unique aspect of our work is that it allows us to study these effects in ground-state molecules which cannot be done using other methods.

We have added the following text at the end of the manuscript on page 14 to clarify this (which also addresses the final comment of Reviewer 2) which reads

Finally, we note that the coherent manipulation experiments and the analysis methods presented in this work rely on the rather general phenomena of the rotational magnetic moment, and are not restricted to H_2 molecules. As such the technique could be used to study both rotationally elastic and inelastic scattering of ground state molecules including HD,

H₂O, NH₃, CH₄ and other small molecules from various metals and insulators, allowing us to obtain empirical scattering matrix benchmarks for a range of systems.

I conclude my comments with an optional suggestion, Following my third question, recent state-to-state scattering experimental and theoretical advances for polyatomic molecules should be referenced, especially if the current method can be extended to study these systems. For example, the Beck group reported vibrationally inelastic state-to-state scattering experiments of CH₄ on Ni(111) [see PRL, 120, 053402] and the Jiang group presented state-to-state quantum scattering calculations for H₂O on Cu(111) [see PRL, 123 106001].

We have added these references to page 13 of the revised manuscript (references 60 and 61).

Reviewer 2

The interaction of hydrogen molecules with solid surfaces has been a model system for understanding gas-surface interactions. Over the years, the experimental measurements and theoretical understanding of this process have become increasingly more sophisticated. The experiments described in this paper take the experimental measurements to the next level.

This is quite well summed up in the first few sentences of their abstract: “The coherent evolution of a molecular quantum state during a molecule-surface collision is a detailed descriptor of the surface interaction potential which was so far inaccessible to measurements. We used a magnetically controlled molecular beam technique to study the collision of rotationally oriented ground state hydrogen molecules with a lithium fluoride (LiF) surface. The coherent quantum control nature of the technique we developed allowed us to measure the changes in the complex amplitudes of all the rotational projection quantum states, and express them using a scattering matrix formalism.”

This is a clearly written and coherent exposition on the new methods being innovated in this research effort. It has many salient points as noted above. I wish to emphasize that the ability to probe experimentally the evolution of molecular wave functions is quite novel, and leads in a straightforward manner to determination of the scattering S-matrix. This is important as it provides a benchmark for theoretical analyses of hydrogen-surface interactions, providing a rigorous test of both interaction potentials and scattering methodologies.

We thank the Reviewer for their supportive comments.

The only point that needs some moderation is the claim that this work is the first to discuss using gas-surface collisions to prepare rotationally polarized or oriented species. This has been discussed previously by several groups in concept based upon molecule-surface scattering experiments that are rotationally analyzed with respect to final quantum state, both for paradigmatic molecular hydrogen interactions in angle-resolved state-to-state scattering experiments as well as laser interrogated quantum-resolved scattering.

It was not our intention to claim that our work is the first to demonstrate that gas-surface collisions rotationally polarised species. We believed we had addressed this point in the original manuscript on page 11 in the paragraph starting ‘Collisions of gas phase molecules with surfaces have previously been demonstrated to create rotational polarisation in scattered molecules’, and through citing work on NO and N₂ scattering from Ag(111) (references 45-47). To further emphasise and clarify this point, we have changed the first sentence of this paragraph on page 11 to explicitly mention a related hydrogen study as follows

Earlier studies have shown that rotationally polarised hydrogen (D_2) molecules can be obtained from the recombinative desorption of D atoms from a Cu(111) surface, where the molecules were found to preferentially desorb rotating like helicopters rather than cartwheels⁴³.

Some key literature citations are clearing missing and need to be added in the proper places and referenced accordingly. Early, seminal work especially for the molecular hydrogen/Ag(111) system (3 references below) must be included as that system provided the best benchmarks based upon combined scattering experiments and extensive quantum scattering calculations encompassing elastic, inelastic, and resonant scattering including some aspects of magnetic sub-level effects. A key paper on hydrogen-graphite also given below, and also needs to be cited. Other papers on scattering from alkali halides could also be added as deeper background if desired.

"Investigation of the Spatially Anisotropic Component of the Laterally Averaged $H_2/Ag(111)$ Physisorption Potential" K.B. Whaley, C.-F. Yu, C.S. Hogg, J.C. Light, and S.J. Sibener J. Chem. Phys. 83, 4235-4255 (1985)

"Investigation of the Spatially Isotropic Component of the Laterally Averaged $H_2/Ag(111)$ Physisorption Potential" C.-F. Yu, K.B. Whaley, C.S. Hogg, and S.J. Sibener J. Chem. Phys. 83, 4217-4234 (1985).

"Selective Adsorption Resonances in the Scattering of $n-H_2$, $p-H_2$, $n-D_2$, and $o-D_2$ from $Ag(111)$ " C.-F. Yu, K.B. Whaley, C.S. Hogg, and S.J. Sibener Phys. Rev. Lett. 51, 2210-2213 (1983).

"Selective adsorption of $1H_2$ and $2H_2$ on the (0001) graphite surface" L. Mattera, F. Rosatelli, C. Salvo, F. Tommasini, U. Valbusa Surf. Sci. 93, 515-525 (1980)

We thank the Reviewer for bringing these references to our attention as examples of rotationally inelastic scattering of H_2 from Ag(111) and graphite surfaces. We have added the first and third reference to the revised manuscript on page 13 (references 62 and 63).

A final comment pertains to the last paragraph of the paper. The authors briefly allude to the generality of their new methods: "Consequently, there is significant potential for using these methods to obtain empirical scattering matrix benchmarks and explore state-to-state stereodynamic effects for a range of ground state molecules colliding with various different surfaces, as well as in gas phase scattering." I think the readership would appreciate some elaboration on this point: What are some other illustrative ground state molecules that these methods can be realistically applied to, what surfaces, and which gas phase systems? What questions can be addressed if these next directions are pursued? Doing this will nicely put into context that these approaches are indeed more general than the test case demonstrated for molecular hydrogen/LiF given herein.

We agree with the Reviewer that these questions should be addressed, and have added the following text on page 14 of the revised manuscript.

Finally, we note that the coherent manipulation experiments and the analysis methods presented in this work rely on the rather general phenomena of the rotational magnetic

moment, and are not restricted to H₂ molecules. As such the technique could be used to study both rotationally elastic and inelastic scattering of ground state molecules such as other isotopes of hydrogen, C₂H₂, H₂O, NH₃, CH₄ and other small molecules from various metals and insulators, allowing us to obtain empirical scattering matrix benchmarks for a wide range of systems.

In summary, this is an excellent and in several aspects pioneering paper. My clear recommendation is to proceed with publication in Nature Communications after the aforementioned straightforward changes and necessary additions are enacted. Well done.

We thank the Reviewer for their enthusiasm for our manuscript and their support.

Reviewer 3

This manuscript reports an experimental study of the scattering of rotationally oriented ground state H₂ molecules from a LiF(100) surface. By using a coherent magnetic field control technique, the authors were able to follow the evolution of molecular quantum wave functions during the collision, which allows performing a complete state-to-state stereodynamics analysis of the H₂ scattering process. A quite remarkable result is the first experimental determination of a scattering matrix, which will provide unique experimental access to a fundamental descriptor of the molecule-surface interaction. Special attention has been paid to analyzing the accuracy and reproducibility of the S-matrix determination, presented in the Supplementary Information. Moreover, these results show that both diffraction channels polarise the rotational orientation of H₂, making it possible the use of LiF(100) as a rotational orientation polarizer.

This is a significant and novel work that will extend the applications of this unique technique to the broad field of stereodynamics. The study has been conducted by one of the leading groups in the field, and will certainly be of great interest to the journal's readership. The article is written clearly and concisely. Subject to what I discuss below, I recommend publication of a suitably revised version of the manuscript in Nature Communications.

We thank the Reviewer for their supportive comments.

- When the current results are compared with previous calculations for the same system performed by Kroes et al., the authors should also mention the first experimental confirmation of large differences in the diffraction of normal- and para-H₂ reported by Bertino and coworkers, Phys. Rev. Lett. 81 (1998) 5608.

We agree with this comment, the reference has been added on page 12 (reference 57).

- Although the observed rotational flip transitions are already present in frozen surface calculations (like the ones reported in Ref. 23), one can expect that surface phonons may influence the flip-mechanism. The authors mention in the Supplementary Information that essentially the same oscillation curves are observed at 165 K and 135 K. I wonder if there is some evidence for a different behavior at higher surface temperatures?

Indeed the Reviewer addresses an interesting point. From the fact that 165 K and 135 K showed similar results it seems likely that assuming a frozen surface at these temperatures might be a valid approach. Unfortunately, we do not yet have data for higher temperatures, having said that considering the significant role that surface temperature can play in sticking

experiments (for example J. Phys. Chem. A, 119, 12434, (2015)), it would be an interesting future study.

- The differences observed for the (1,0) and (-1,0) diffraction peaks (Fig.3a,b) are quite clear, in part due to the small error bars. I understand that the small error bars correspond to the small uncertainty when measuring the diffraction intensities. I think this information is relevant and should be added to the main text.

The error bars reflect the uncertainties from repeat measurements of the experimental data. We have added a sentence to the main text at the top of page 8 which states that

These are the average of at least five identical B1 scans, with the error bars reflecting the uncertainty in the data estimated from the scatter of the measured values.

- Second order diffraction is also observed from this surface (see paper by Bertino et al. mentioned above). I guess the current work is focused on the first order diffraction peaks, (1,0) and (-1,0), due to their higher intensity. However, it would be interesting to discuss what kind of additional information could be obtained by performing a similar analysis to second order diffraction peaks, which in principle should be possible using the authors' experimental setup.

The Reviewer is correct that we focussed on the first order diffraction peaks due to their higher intensity, but it would be possible to perform the experiments on higher order diffraction peaks. We would expect that the different diffraction peaks would display different rotational orientation polarisation properties in the same way as these are different for the (1,0) and (-1,0) diffraction peaks, and hope to perform such measurements in the future.

REVIEWERS' COMMENTS:

Reviewer #1 (Remarks to the Author):

I am satisfied with the authors' response and will now recommend the publication of this work.

Reviewer 1

I am satisfied with the authors' response and will now recommend the publication of this work.

We thank the Reviewer for their support.